# Planar Installation Characteristics of Crown Depth-Variable Artificial Coral Reef on Improving Coastal Resilience: A 3D Large-Scale Experiment

Sunghoon Hong [1] , Seungil Baek [2] , Yeonjoong Kim [3] , Jooyong Lee [1], Adi Prasetyo [4], Wonkook Kim [2] and Soonchul Kwon [2,*]

1   Research Institute of Industrial Technology, Pusan National University, Busan 46241, Korea; superkick701@naver.com (S.H.); jooyong@gmail.com (J.L.)
2   Department of Civil and Environmental Engineering, Pusan National University, Busan 46241, Korea; seung1100@pusan.ac.kr (S.B.); wonkook@pusan.ac.kr (W.K.)
3   Department of Civil and Urban Engineering, Inje University, Gimhae 50834, Korea; anyseason@inje.ac.kr
4   Experimental Station for Coastal Engineering, Directorate General of Water Resources, Ministry of Public Works and Housing, Bali 81155, Indonesia; adi.prasetyo246@yahoo.com
*   Correspondence: sckwon@pusan.ac.kr

**Abstract:** Coastal resilience has received significant attention for managing beach erosion issues. We introduced flexible artificial coral reef (ACR) structures to diminish coastal erosion, but planar installation effects should be considered to evaluate the feasibility of coastline maintenance. In this study, we conducted a three-dimensional large-scale experiment to investigate the characteristics of planar installation of ACR, focusing on the wave mitigation performance, wave profile deformation with delay, nearshore current movement, deposition and erosion trends, and beach profile variation. We found that the ACR diminished the wave height by ~50% and the current intensity by ~60% compared with that of a conventional submerged breakwater made of dolos units. Using the dispersion velocity of the dye in a tracer experiment, the dispersion time of the ACR was approximately 1.67-times longer than that of the dolos and the current velocity was reduced, revealing that ACR significantly reduced structural erosion. With dolos, severe erosion of >10 cm occurred behind the structure, whereas there was only slight erosion with the ACR. Moreover, in a vertical beach-profile analysis, the ACR exhibited greater shoreline accretion than that of dolos. These results indicate the potential of ACR in improving coastal resilience.

**Keywords:** artificial coral reef; coastal resilience; wave mitigation; nearshore current; topographical trend

## 1. Introduction

### 1.1. Coastal Erosion Prevention Method and 'Coastal Resilience' Concept

According to long-term shoreline monitoring using satellites from 1984 to 2015, gradually worsening coastal erosion has caused the loss of permanent land (28,000 km$^2$), which accounts for more than double the deposition area [1]. Moreover, rapid and severe erosion of more than 0.5 mm/year has continuously occurred on sandy beaches worldwide [2]. The main reported reasons for coastal erosion are rising sea levels, abnormal climates due to global warming [3,4], side effects of manmade coastal structures [5,6], and land subsidence [7], all of which threaten coastline stability. To address these erosion issues, numerous studies have been conducted with various analytical approaches, such as hydraulic experiments, numerical modeling, and field monitoring to evaluate coastal prevention methods.

Using a large number (>2300) of previously obtained hydraulic experiment data sets, Van der Meer et al. [8] investigated the wave transmission and reflection of rubble mound low-crested structures (LCS) based on the relative freeboard, relative crest width, wave

direction and spectral change, porosity, and breakwater parameter, which represent important factors of offshore coastal structures. Hur et al. [9] examined the effects of various types of submerged breakwaters with drainage channels on the wave attenuation, wave profile variation, and mean water level reduction to mitigate nearshore currents. Cappietti et al. [10] analyzed field wave data to evaluate prior empirical models of transmission and setup and considered their reliability based on root mean square (RMS) error values. To investigate the current movement around an artificial reef, Osanai and Minami [11] measured the vertical and horizontal average velocity with a three-dimensional acoustic Doppler velocimeter (3D ADV) and found dominant onshore and downward current directions based on current vector results. Lee et al. [12] examined the effects of the beachface slope in terms of the mean wave height trends, average flow rate, and wave breaking points by performing a numerical analysis with the LES-WASS-3D model. With regard to topographical analyses, Alesheikh et al. [13] detected coastline variation through aerial photography using airborne vehicles or unmanned aerial vehicles, and Burns et al. [14] used aerial light detection and ranging (LiDAR) survey techniques and analyzed elevation changes by producing highly accurate 3D models. These studies demonstrated that remote sensing techniques can be used to quantify shoreline variations.

Based on prior studies, many countries have attempted to restore and maintain their coasts through large-scale national projects, such as the "Sand Motor Project" [15] of the Netherlands and "Coastal Maintenance Project" [16] of South Korea. However, in some cases, conventional coastal protection methods have failed despite enormous budgets and time expanses [17]. These results stress the need for long-term sustainable coastal environment strategies to fundamentally manage coastal erosion problems. In this regard, the European Commission introduced the concept of "coastal resilience", which addresses adaptability and recovery functions from threats, such as disasters in coastal erosion management [18]. Moreover, Masselink and Lazarus [19] defined the coastal resilience concept by the "capacity of natural systems to cope with disturbances (sea level rise, extreme events, human impacts) by maintaining their essential functions" and postulated that the ecological component for nature conservation, such as coral reefs, salt marshes, mangroves, and ETC, could be fundamental solution for enhancing coastal resilience.

*1.2. Artificial Coral Reef (ACR) Structures*

To enhance coastal resilience with ecological components, we focused on the natural coral reef, which is known for its functions in maintaining a stable coastline. In the case of Hainan Island (China), severe erosion has occurred in Puqian Bay due to coral reef deterioration; however, the coastline has remained stable for 20 years with healthy coral reef colonies [20]. Reguero et al. [21] investigated the shoreline protection functions of coral reefs based on historical shoreline data and used a "beach equilibrium planform model", demonstrating the positive effects of a coral reef on shoreline stability. Moreover, Huang et al. [22] discussed the correlation between wave damping, and turbulent kinetic energy increment due to the coral reef–lagoon interaction and drag force effect of the coral head. By considering the erosion control functions of the coral reef, Harris [23] investigated the beach-stabilizing effects of reef ball type artificial reefs based on shoreline and sand volume data. Buccino et al. [24] reanalyzed >300 experimental datasets of a conventional submerged breakwater to develop a set of equations for reef balls (RBs) on the wave transmission coefficient, which is regarded as a useful indicator of wave energy dissipation. In addition, the results of various experiments showed that the degree of submergence, configuration factor, and permeability of the RB can significantly affect the transmission coefficient. Srisuwan and Rattanamanee [25] investigated the surface wave attenuation performance of a seadome by analyzing >2600 laboratory experiment results and developed empirical formulas. The results of the comparison analysis demonstrated a reliable correspondence between the laboratory data and modeled results, which could provide a practical solution for designing seadome structures. Gourlay [26] examined the wave setup effects, owing to an idealized horizontal coral reef by employing hydraulic

experiments, which showed that the incident wave height, wave period, and water level are key parameters for both the wave setup and wave-generated flow. In addition, the results of three specific reef sites in terms of wave setup demonstrated the reliability of "wave set-up" prediction [27].

In this study, we introduce an artificial coral reef (ACR) as a coastal prevention method. The ACR resembles the shape of a natural coral reef, consisting of high-density polyethylene (HDPE). The ACR unit is designed with an oval cylinder-shaped "sand trap" for drift sand deposition to its inner space and a "wave trap" for wave attenuation by roughness effects and wave breaking, which contribute to the mitigation of beach erosion (Figure 1a). According to a preliminary experiment, the deposition capacity for each sand trap is determined by the column height. For this reason, additional sedimentation does not occur when the sand trap reaches its deposition capacity. Even though the sand trap exceeds its capacity, the deposited area due to the ACR can reduce the water depth and form a beach profile with a mild and gradual slope, which is effective for preventing drastic wave breaking.

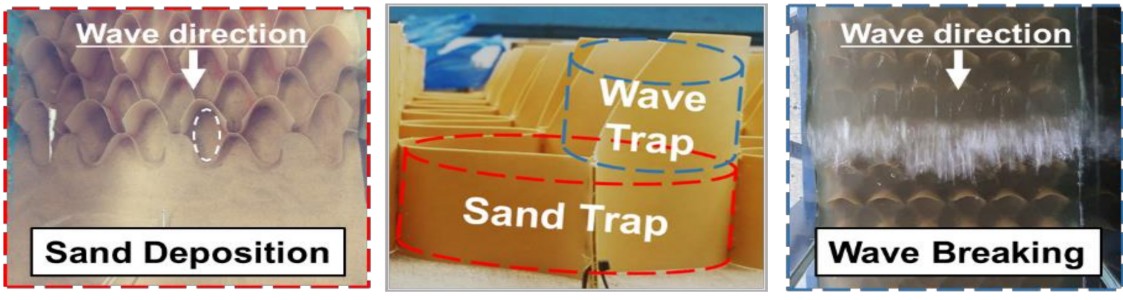

(**a**) Structural parts and functions of the ACR
(the white dashed ellipse represents the inner space for sedimentation)

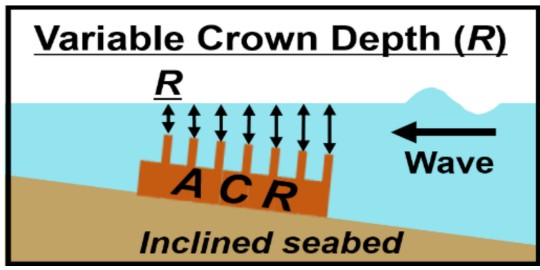

(**b**) Geometry of the ACR

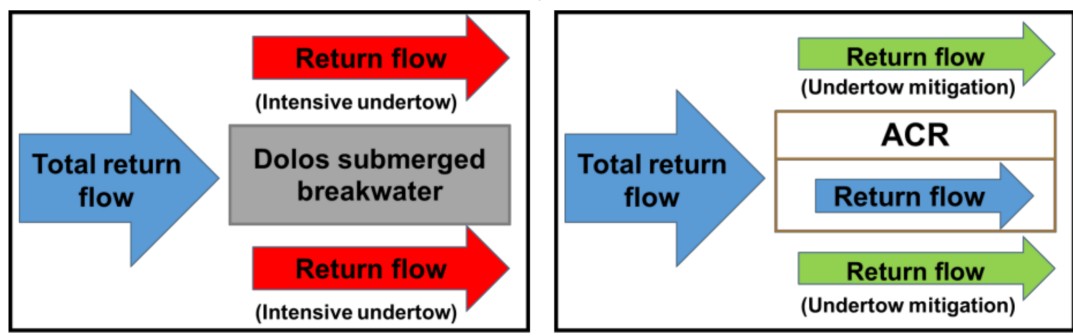

(**c**) Undertow mitigation scheme with return flow distribution

**Figure 1.** Structural parts and functions (**a**), geometry (**b**) and undertow mitigation (**c**) scheme of the ACR.

The group of ACR units have geometries with large voids and gradually variable crown depths that lead to a large porosity (Figure 1b). According to the results of prior studies [28–30], submerged structures with large porosities tend to exhibit large transmis-

sion and small reflection effects and allow smooth fluid exchange through their structural units. For this reason, an ACR has advantageous functions in preventing rapid undertow mitigation by facilitating fluid exchange through its structural unit (Figure 1c).

In prior ACR studies, the analysis of the wave reduction performance and erosion mitigation effects for improving coastal resilience were primarily discussed. Hong et al. [31] conducted a two-dimensional hydraulic experiment and examined both the wave reduction performance, shoreline variation, and erosion mitigation effects. The ratio of deposited sand volume in the ACR to total beach sand erosion (efficiency of beach sand) and the wave transmission coefficient were ~24% and 0.74, respectively, which play a positive role in shoreline stability. Jeong [32] addressed both wave and current control functions and initial sediment transport around the ACR by developing a numerical wave tank–discrete element method (NWT–DEM) two-way coupled model, which revealed the efficiency of the ACR on coastal prevention as a soft defense method. Moreover, Hong et al. [33] simulated and analyzed the application of ACR in the rear sea by employing a combined analysis, based on a simulating waves nearshore (SWAN) numerical model and physical model experiment. The trends of wave and current control and decrement of two indicators (Dean's and Surf-scaling parameters) demonstrated the potential of ACR in terms of shoreline protection.

In general, time and cost limitations restrict the dimensions of a coastal prevention structure, which suggests that understanding the planar effects around an ACR is essential for its practical application in the coastal zone. Despite its importance, prior studies have not addressed planar effects, such as wave deformation, current behavior, and beach morphology. Therefore, in this study we carried out a 3D physical model experiment and conducted a comparative analysis between an ACR and dolos submerged breakwater to elucidate both the hydraulic characteristics and topographical trends resulting from the planar installation of the ACR. The specific aims of this study were to: (1) examine the characteristics of wave mitigation and wave profile deformation with wave delay; (2) investigate the movement of the nearshore current and distribution of the current vector; and (3) reveal the erosion and sedimentation trends, which can validate the planar effects of an ACR on coastal resilience enhancement.

## 2. Materials and Methods

To identify the planar installation characteristics of an ACR, we conducted a 3D physical model experiment (see Figure A1) with a movable bed. During the experiment, we conducted a comparative analysis between an ACR and a conventional submerged breakwater made of a dolos armor unit. We selected dolos as the reference structure because it has representative properties; it is made of a rigid material and small voids, which can emphasize the characteristics of the ACR. The following sections describe the specifications of the research facility, methodology, and theoretical background of this study.

### 2.1. Sandwork and Physical Model Construction

When a sand particle is too large in a movable-bed experiment, the total time for topographical change can be excessive. For this reason, we used a large sized sieve to obtain filtered sand with median grain sizes ($D_{50}$) of 0.2 mm (see Figure A2). The beach was made of filtered and non-filtered sand ($D_{50} = 0.3–0.4$ mm) layers. First, we laid non-filtered sand on the basin floor, where no interaction occurred between the sand layer and wave. We made a 1/20 slope with a filtered sand layer on the non-filtered sand to obtain a more precise topographical variation. We used the segmented wave generator with paddle type mode (Rexroth Bosch Group, Lohr am Main, Germany), owned by the "Balai Teknik Pantai, ministry of public works and housing (Bali, Indonesia)", to conduct a physical model experiment (Figure A3a). Moreover, we constructed a wave basin with dimensions 30 m (length) × 12 m (width) × 1 m (height) for this study (Figure A3b).

As the dominant factors of water waves are inertia and gravitational forces, we applied the Froude similarity for the physical model. For this reason, the length and time scale ratio

for the model to prototype were 1:25 and 1:5, respectively, based on the constraints of both the wave basin size and wave generator specification. Figure 2 shows the dimensions of the dolos and ACR. We downsized the length of both structures to 6 m (150 m in proto), which were located in the middle of the wave basin so that both structures had 3 m length open inlets on the left and right sides. For this reason, we reviewed the boundary effect of the breakwater gap on wave diffraction at the open inlet location. According to experimental results analyzed by Pos and Kilner [34], for the following ratios: breakwater gap to wave length of 1.41, displacement (2 m) between the basin wall and open inlet to the wave length (2 m) of 1, and offshore distance (2.8 m and 2 m) to wavelength (2 m) is in the range of 1–1.5, the wave diffraction coefficient is expected to be in the range 0.6–0.7, which is relatively reliable for the physical model experiment.

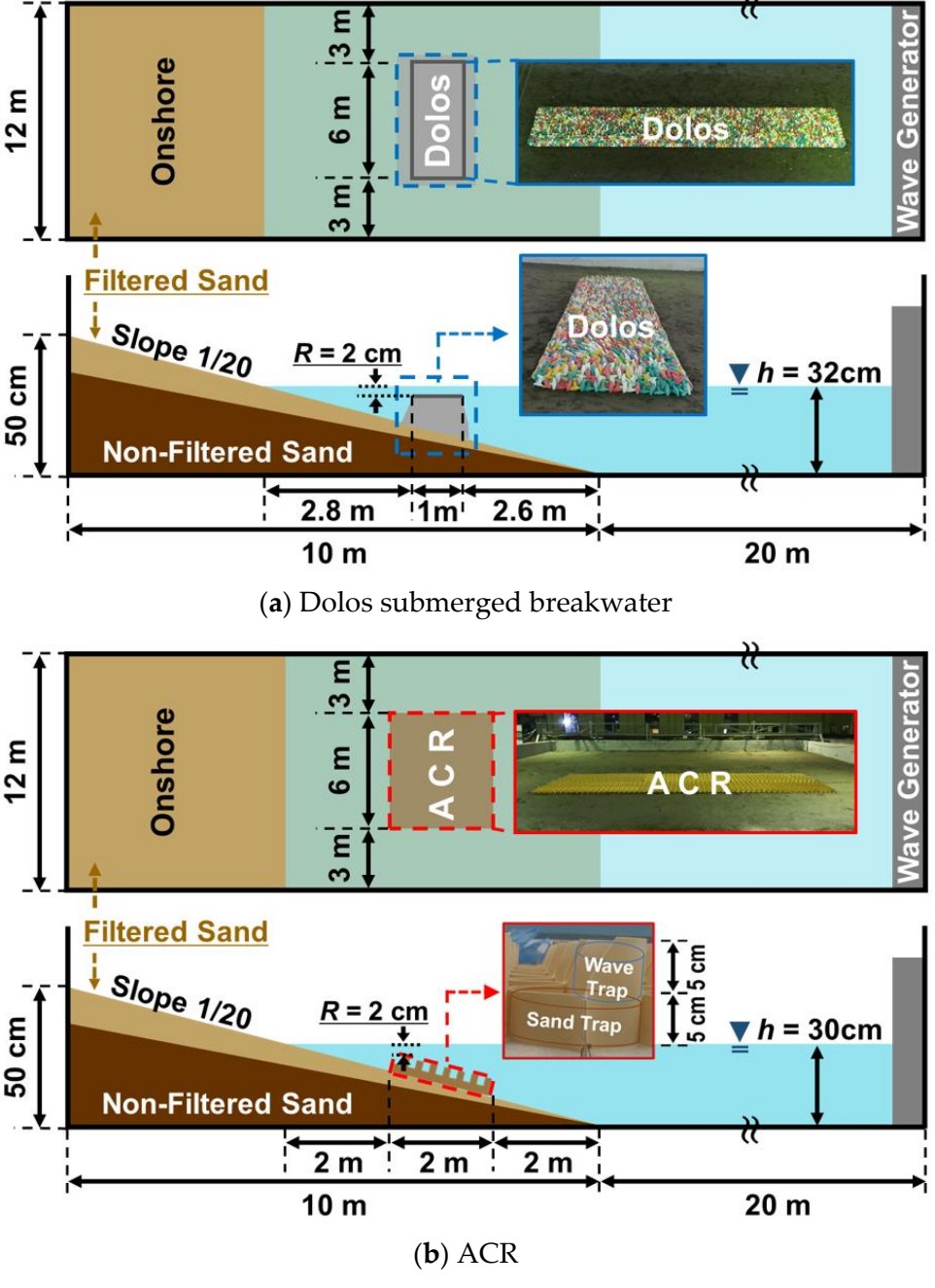

(**a**) Dolos submerged breakwater

(**b**) ACR

**Figure 2.** Dimensions of the experimental test bed with dolos (**a**) and ACR (**b**) (the vertical scale is exaggerated to improve visibility).

In addition, we designed the ACR to have a doubled crest width (2 m) compared to that of the dolos (1 m), to compensate for its mild wave control performance owing to the variable crown depth. The water depths of the dolos and ACR were 32 and 30 cm, respectively. The crown depth ($R$), which is the vertical distance between the structure head and water surface, is an important factor affecting wave attenuation; therefore, we referred to the general crown depth design (0.5 m) of submerged breakwater and set the crown depth to 2 cm for both structures.

## 2.2. Wave Condition for the Physical Model Experiment

To investigate the erosion mitigation effects of the ACR under high waves, we assumed a significant wave with 2.5 m height ($H_o$) and 6.5 s period ($T_o$) for the movable-bed physical model experiment. In this regard, we introduced Dean's parameter [35], which indicates the beach state, such as erosion or deposition, to review the suitability of the experimental wave condition. Dean's parameter is a function of the wave-breaking height ($H_b$), settling velocity of sand particles ($w_f$), wave period ($T_i$), and wave-breaking depth ($h_b$). We performed the computation procedures as follows.

First, we built a numerical wave basin, which has the same dimensions as the present wave basin and applied the SWAN third generation model [36] to determine the wave-breaking height ($H_b$). In addition, we introduced the concept of the breaker index ($\kappa$, general value of 0.78), suggested by McCowan [37], to calculate the theoretical maximum wave height based on the water depth, as shown in Equation (1). After this, we determined both the height ($H_b$) and depth ($h_b$) as 7.1 and 9 cm, respectively, by comparing the wave height value based on the SWAN modelling and breaker index computation (Figure A4).

$$H_b = \kappa \times h_b = 0.78 \times h_b \tag{1}$$

We used filtered sand with a median grain size ($D_{50}$) of 0.2 mm, and estimated the settling velocity ($w_f$) based on the formula of Zanke [38]. Note that the median grain size is in the range $0.1 < D_{50} \leq 1.0$ mm.

$$w_f = \frac{10\nu}{D_{50}} \times \left[ \left\{ 1 + \frac{0.01(s-1)g(D_{50})^3}{\nu^2} \right\}^{0.5} - 1 \right] = 2.57 \text{ cm/s} \tag{2}$$

where $\nu$, $s$, $g$, and $D_{50}$ represent the kinematic viscosity of the water ($1 \times 10^{-6}$ m$^2$/s), specific weight of the sand particle (2.65), acceleration of gravity (9.81 m/s$^2$), and median grain size, respectively.

With the specific values in the model scale, we computed the dimensionless Dean's parameter ($\Omega$), as given by Equation (3).

$$\Omega = \frac{H_b}{w_f \times T_i} = \frac{7.1 \text{ cm}}{2.57 \text{ cm/s} \times 1.3 \text{ s}} = 2.13 \text{ (intermediate beach state)} \tag{3}$$

where $T_i$ is the wave period condition in the model scale (6.5/5 s = 1.3 s).

According to Wright and Short [39], we computed the result of Dean's parameter ($\Omega = 2.13$) corresponding to an 'intermediate beach state', representing a value between 'dissipative ($3.0 < \Omega$)' and 'reflective ($\Omega < 1.0$)' beach conditions. For this reason, the wave conditions in the experiment are appropriate for analyzing the topographical trend results obtained with both the dolos and ACR.

## 2.3. Wave Measurement and Analysis

We used a wave gauge and four sets of wave probes (HR Wallingford, Wallingford, UK) to examine wave mitigation effects and wave profile deformation. Figure 3 shows the locations of the wave probes. The vertical locations for the wave probe were set at 3 cm beneath the water surface and 1 cm at the dolos crest.

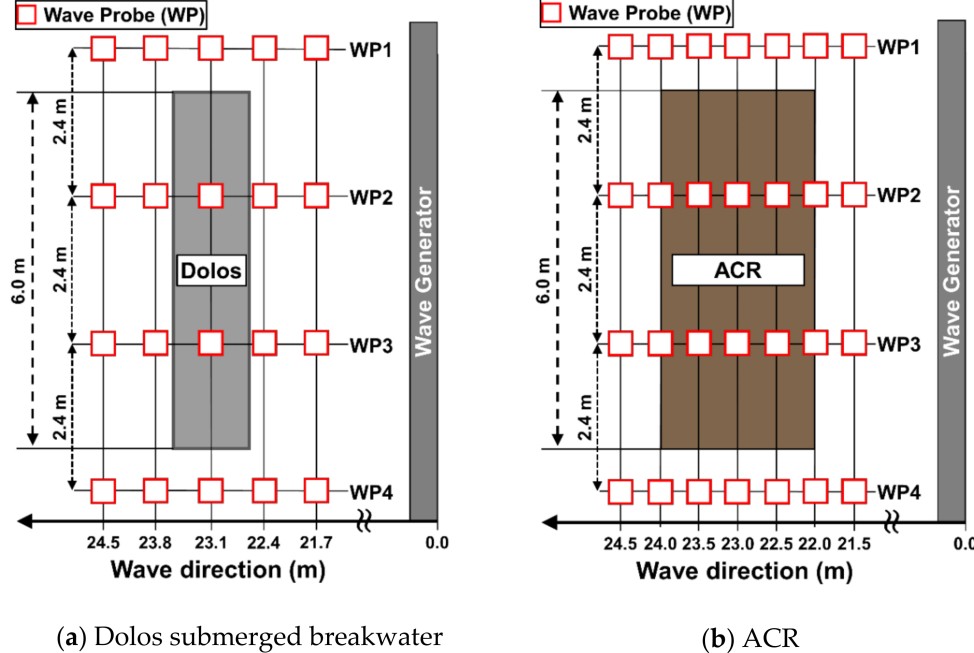

(**a**) Dolos submerged breakwater        (**b**) ACR

**Figure 3.** Wave measurement location with dolos (**a**) and ACR (**b**) (WP: Wave Probe).

To ensure wave data reliability, we conducted wave probe calibrations based on the linear relationship between the water level and voltage (Figure A5) at all points and measured 230 waves for 3 min, sufficient for a wave analysis [40]. We set the incident wave (incident wave height: $H_i$ and $T_i$) and applied the representative wave theory and zero-up crossing method to conduct the wave analysis. Moreover, we computed the attenuation rate ($R_A$), which is the ratio of the wave height decrease ($H_i - H$) to $H_i$, to quantitatively evaluate the wave attenuation performance:

$$R_A = \frac{H_i - H}{H_i}. \tag{4}$$

We investigated the wave deformation trends that result from offshore structures. In this respect, we sampled three consecutive wave trains ($t = 3T_i$) and conducted a qualitative analysis of the wave profile ($\eta$) variation. During the wave profile analysis, we applied the concepts of kurtosis and skewness, i.e., vertical and horizontal asymmetry, respectively, in the same manner as in a previous study [41]. In addition, we defined the wave delay parameter ($\Delta T$) to compute the time differences between the structure absence ($T_A$) and presence ($T_P$) locations (Equation (5)).

$$\Delta T = T_A - T_P \tag{5}$$

$$\overline{\Delta T} = \frac{(\Delta T_1 + \Delta T_2 + \Delta T_3)}{3} \tag{6}$$

where, $T_A$ and $T_P$ represent the elapsed time at the WP4 and WP3 locations, respectively.

Note that we set the wave crest as a reference point for the $\Delta T$ computation because it has one prominent maximum point (see Figure 4). A positive $\Delta T$ indicates that the wave at WP3 propagates earlier than at WP4 (refer to Figure 3 for each location). In contrast, a negative $\Delta T$ means that the waves at WP3 propagate more slowly than those at WP4. To improve the data reliability of the wave delay, we introduced a parameter $\overline{\Delta T}$, termed the averaged wave delay parameter, which represents the averaged delay of three consecutive and different wave trains ($\Delta T_1$, $\Delta T_2$, and $\Delta T_3$) as shown in Equation (6) (Figure 4).

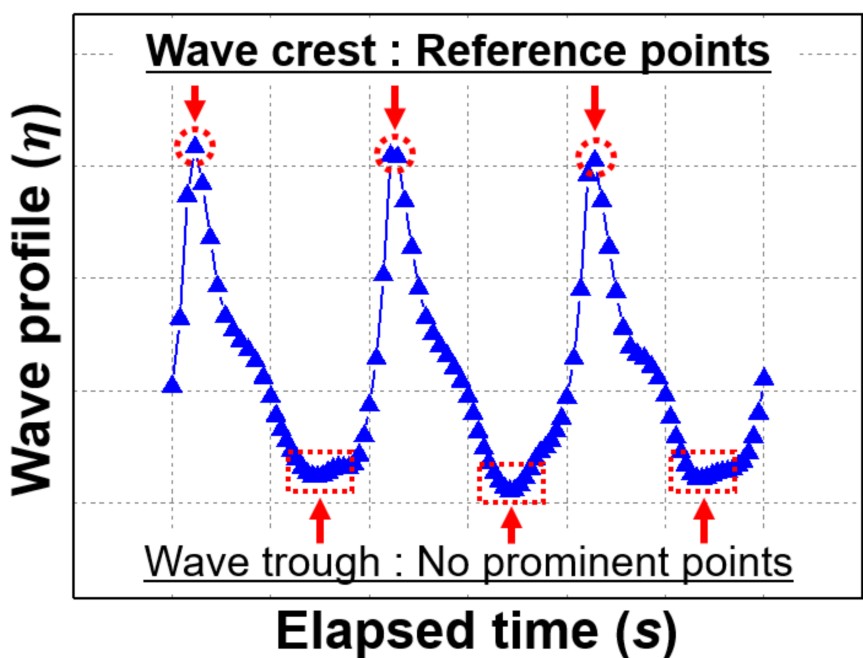

**Figure 4.** Reference time determination scheme.

### 2.4. Analytic Methods for Nearshore Current

To understand the behavior of nearshore currents around the dolos and ACR, we performed both qualitative and quantitative analyses. For the qualitative analysis, we released a water-soluble red dye (Rhodamine B) and recorded video using a drone (Phantom, DJI, Shenzhen, China) to trace its path. During the aerial image analysis, we assumed a current velocity based on the displacement of the dye and dispersion time from the initial release point to the offshore boundary of each structure.

For the quantitative approaches, we measured both the horizontal velocity (along the *x*-axis) and vertical velocity (along the *z*-axis), *u* and *w*, respectively, around the structure using a high-resolution Doppler current meter (Micro ADV, SonTek, Inc., San Diego, CA, USA). Positive values (+) were set as the wave direction and vertical uplift direction for the horizontal and vertical velocities, respectively. Figure 5 shows the locations of the ADV sensors in the breakwater middle (BM), breakwater shoulder (BS), and breakwater open inlet (BO). For each location, we measured both the horizontal and vertical current velocities from the seabed to the water surface at intervals of 5 cm.

To clarify the domain current property at each point, we computed the average values of the horizontal and vertical currents as follows:

$$\overline{u} = \frac{1}{N} \times \sum_{1}^{N} u_i \, , \ \overline{w} = \frac{1}{N} \times \sum_{1}^{N} w_i, \tag{7}$$

$$\overline{V} = \sqrt{\overline{u}^2 + \overline{w}^2}, \ \theta = \mathrm{ACOS}\left(\frac{\overline{u}}{\overline{V}}\right) \times \frac{360°}{2\pi} \tag{8}$$

where *N* represents the number of current data points. We determined the magnitude and direction of the current vector according to the computed horizontal and vertical current components to investigate the current properties. We used only current data with signal-to-noise ratios >30 to maintain data reliability.

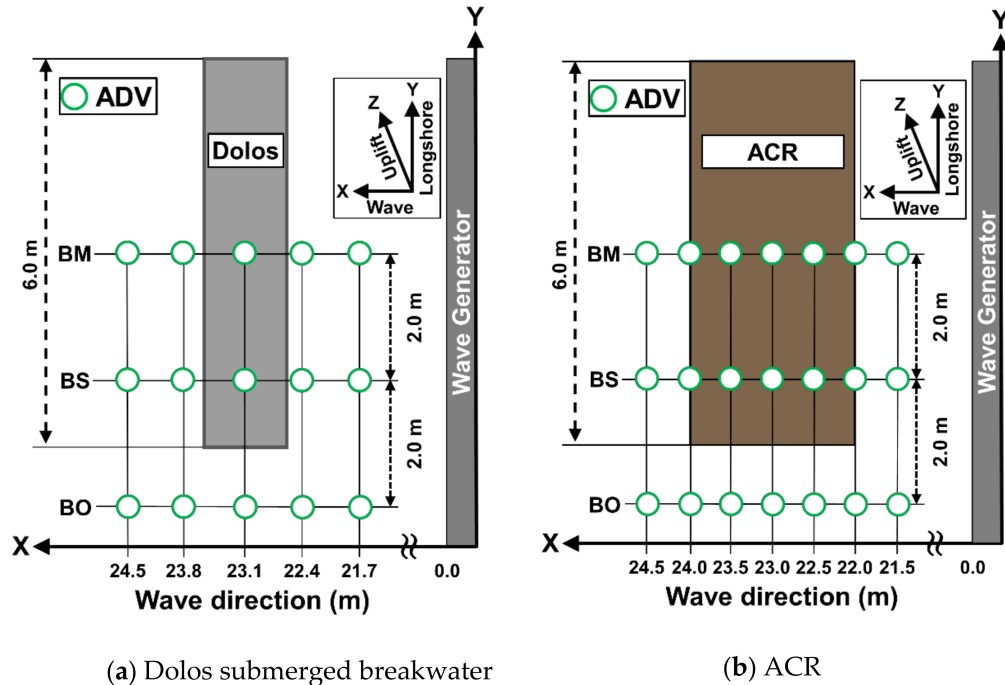

(**a**) Dolos submerged breakwater      (**b**) ACR

**Figure 5.** Current measurement locations with the ADV sensor (dolos—(**a**) and ACR (**b**)).

### 2.5. Topographical Variation Tendency Analysis

A LiDAR scanner (GLS-1500, Topcon Corp., Tokyo, Japan) was used to obtain elevation data to investigate topographical variation trends (Figure 6a). As the LiDAR scanner acquires point-based vector data, rasterization is necessary. Therefore, we converted the spatial coordinate data into a rasterized digital elevation model (DEM) with a spectral resolution of 1 cm. Owing to the straightforward property of the laser, the elevation data acquisition was limited, particularly at the rear sides of both structures. For this reason, we located the LiDAR scanner on the left and right of both structures, as well as the front, and superimposed the elevation data to deal with shadow zone issues. Moreover, to investigate erosion and deposition trends around the structure, we defined the difference of DEM (DoD), which refers to the amount of vertical variation between the initial (0 h) and final (20 h) topography (Figure 6b). In this regard, positive and negative DoD values indicate sedimentation and erosion, respectively, for each location.

In addition to conducting a comparative analysis between the dolos and ACR, we extracted three representative beach profiles, which are located in the BM, BS, and BO, as mentioned above. For an efficient analysis, we applied a smoothing procedure with a Savitzky–Golay filter to reduce the elevation data noise (Figure 6c).

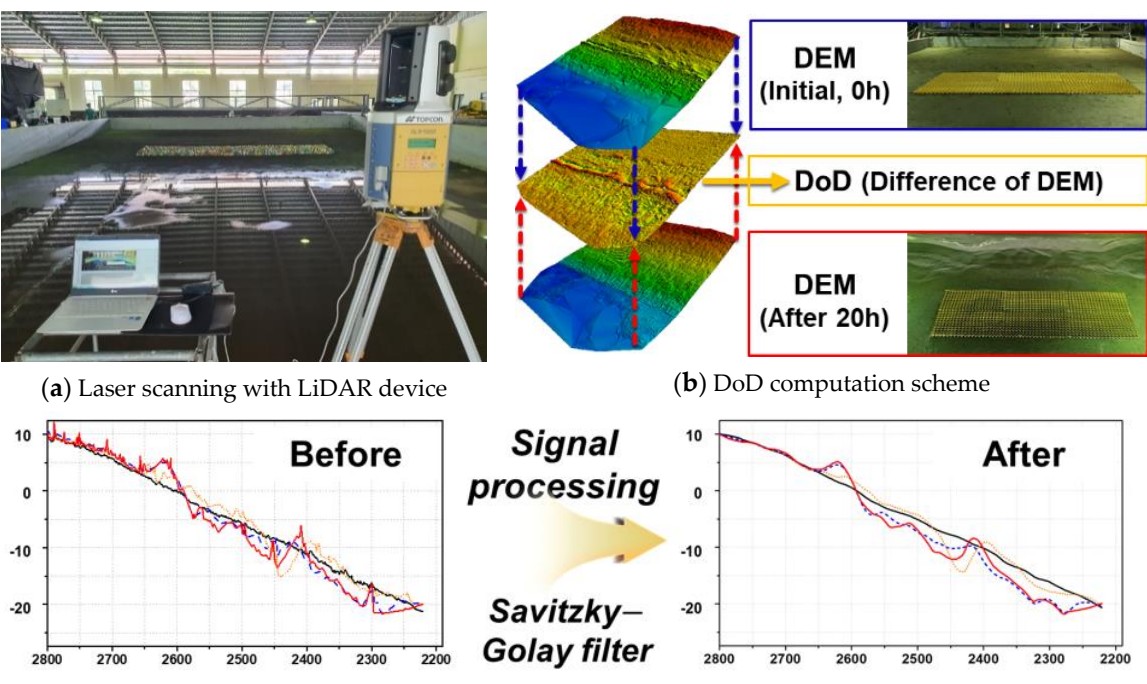

(**a**) Laser scanning with LiDAR device      (**b**) DoD computation scheme

(**c**) Signal process for removing DEM data noise

**Figure 6.** Topography analysis methods with LiDAR device (**a**,**b**) and signal processing (**c**).

### 3. Results and Discussion

*3.1. Wave Mitigation Effects and Wave Profile Deformation Trends*

To investigate the wave attenuation trends of the ACR, we computed the attenuation rate ($R_A$) around the dolos and ACR structures (Figure 7 and Appendix A Table A1). Note that the *x*-axis in Figure 7 represents the dimensionless relative distance ($X/L_O$), which is the displacement ($X$) to 2 m of wavelength ($L_O$). First, the average $R_A$ values of the outer (WP1 and WP4) and inner (WP2 and WP3) domains were 0.155 and 0.535 for the dolos, and 0.230 and 0.460 for the ACR, respectively.

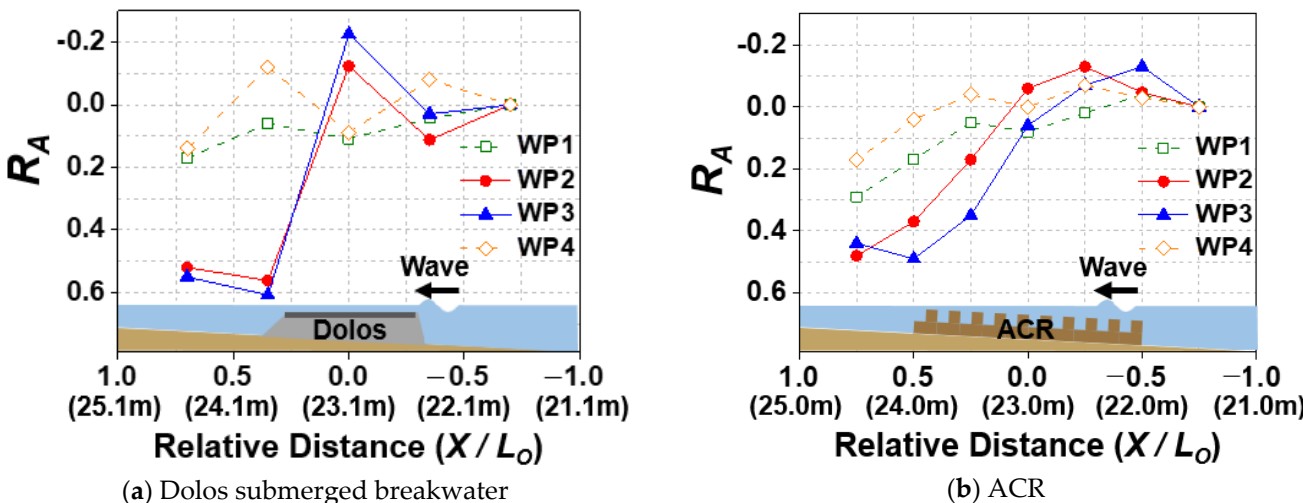

(**a**) Dolos submerged breakwater      (**b**) ACR

**Figure 7.** Attenuation rate ($R_A$) results (dolos—(**a**) and ACR—(**b**)).

In addition, the wave mitigation effects of the dolos slightly exceeded the ACR of 0.075 at the rear sides of both structures. However, there was also a difference in $R_A$ between the outer and inner domains: for the dolos $\Delta R_A = 0.38$ and ACR ($\Delta R_A = 0.23$), particularly at the onshore boundary for each structure.

As the water level gradient is the main reason for current generation, drastic water level variations induced strong nearshore currents around the dolos, which caused negative phenomena, such as scour and erosion: this was particularly observed at the rear side of the onshore boundary structure. On the contrary, a mild water level variation occurred in the ACR. These results demonstrate that the advantageous structural properties of ACRs, such as a variable crown depth and large porosity, induce mild wave deformation, which plays a positive role in shoreline protection. In summary, compared with the dolos, an ACR has gradual wave-mitigating effects due to its structural properties, such as variable crown height and large voids. For this reason, an ACR may be effective in terms of nearshore current mitigation, particularly in offshore structures.

To examine the wave deformation property of the ACR, we analyzed the wave profile ($\eta$) data at the incident wave, structure center, and transmitted wave locations. Figure 8 shows the wave profiles of WP3 (inner domain) and WP4 (outer domain) for both the dolos and ACR cases during the three wave trains.

At the incident wave location, the wave profiles of both WP3 and WP4 were symmetric without any deformation (Figure 8a,b). When the waves propagated over the dolos center location, rapid wave height increments and decrements occurred in WP3 and WP4, respectively (Figure 8c). In contrast, no prominent wave profile variation occurred for WP3 and WP4 at the ACR center location (Figure 8d).

After the waves passed over the dolos, the vertical fluctuation decreased by ~60% for WP3 and ~20% for WP4 compared to that of the initial wave profile (Figure 8e). In addition, the skewness and kurtosis of the wave profile significantly increased, which means that the asymmetry of both the horizontal and vertical directions increased owing to the strong and direct interaction between the waves and dolos. Consequently, the regular wave profile changed into a nonlinear shape. In contrast, the wave profile of the ACR had a relatively regular shape compared to that of the dolos cases (Figure 8f). For instance, the wave profile variation diminished by only ~48% and ~5% for WP3 and WP4, respectively, when comparing the incident wave profiles. Moreover, the skewness and kurtosis of the ACR seem smaller than those of the dolos, which means that no rapid wave deformation occurred because of the ACR structure.

During the wave profile analysis, we observed a wave delay phenomenon between WP3 (inner domain) and WP4 (outer domain) for both the dolos and ACR cases. According to prior studies [42], the interaction between water and coastal structures causes the wave delay phenomenon. In this respect, Table 1 shows the results of the averaged wave delay parameters ($\overline{\Delta T}$) at each location. Firstly, at the structure's center location, $\overline{\Delta T}$ of the dolos increased to 0.08 s, whereas no wave delay occurred in the ACR case. This is because the rapid cross-sectional area decrement contributed to an increase in the current velocity for the dolos case. Secondly, in the transmitted wave location, $\overline{\Delta T}$ of both the dolos and ACR decreased to −0.15 and −0.09 s, respectively, compared to those at the incident wave location. The $\overline{\Delta T}$ results for the dolos case showed a 1.67 times greater wave delay effect than that of the ACR case. In addition, the relatively large $\overline{\Delta T}$ of the dolos represents the possibility of high wave diffraction behind its structure. In contrast, the wave trains at WP3 and WP4 propagated simultaneously for the ACR. In summary, compared to the conventional dolos submerged breakwater, the ACR exhibited mild and gradual wave deformation trends, which is advantageous in terms of wave diffraction and nearshore current control around its structure.

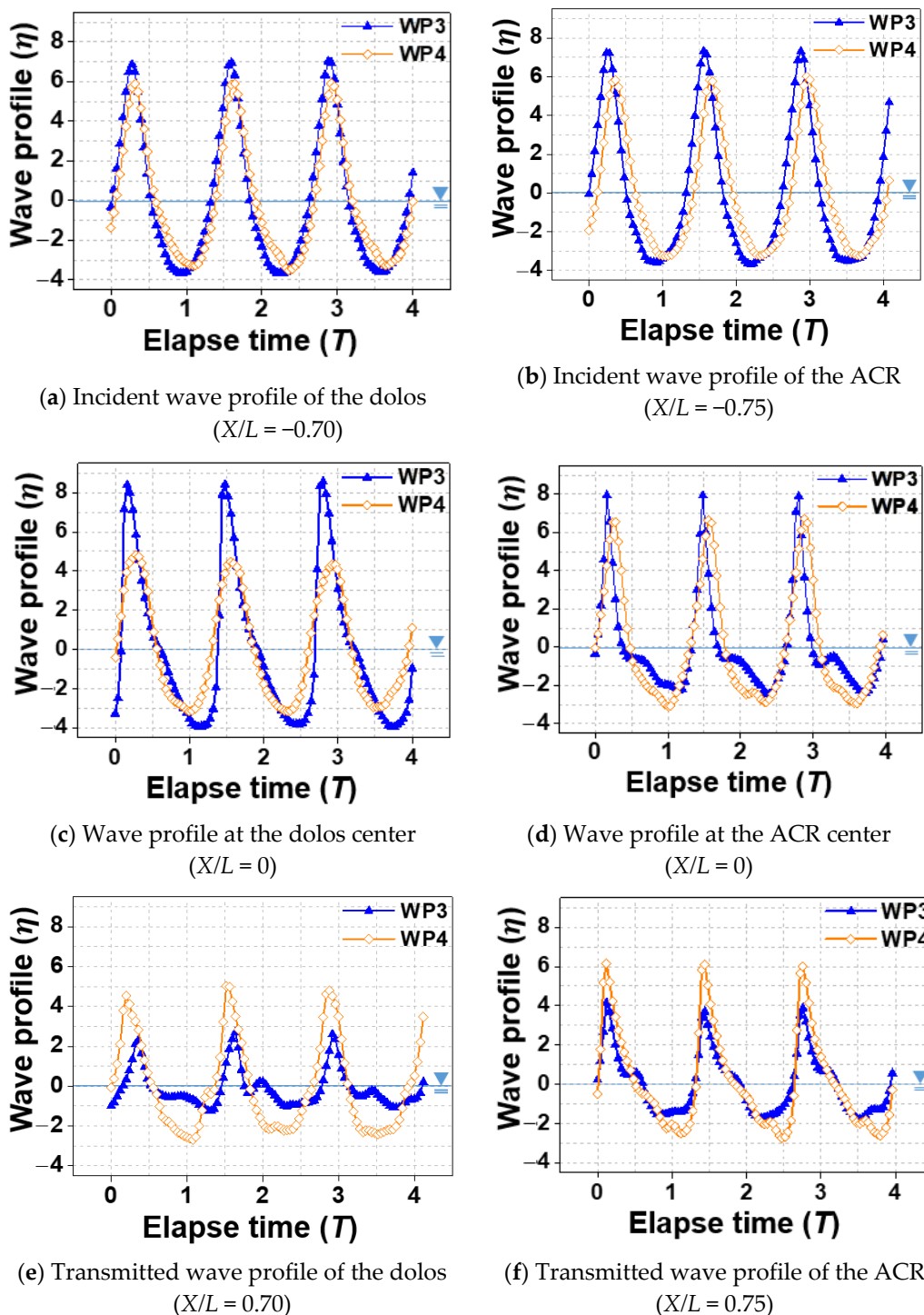

**Figure 8.** Wave profile deformation with phase delay around dolos (**a**,**c**,**e**) and ACR (**b**,**d**,**f**).

To elucidate the wave profile deformation and phase delay, we analyzed the time series data of the transmitted waves of both structures and plotted the frequency spectrum (Figure 9). In the dolos case, we observed a prominent energy distribution from the spectrum barycenter to the lower and higher frequency domains owing to the interaction between the propagating and reflected waves. In contrast, the concentration of spectral energy showed a particularly abundant peak frequency, which means that no significant wave deformation occurred during wave propagation over the ACR. According to these results, the ACR exhibits a different wave control mechanism compared to the conventional submerged breakwater.

**Table 1.** Wave delay parameter results (in units of s).

| Location | Dolos | | | ACR | | |
|---|---|---|---|---|---|---|
| Incident wave | $\Delta T_1 = 0.04$ | $\Delta T_2 = 0.04$ | $\Delta T_3 = 0.04$ | $\Delta T_1 = 0.12$ | $\Delta T_2 = 0.08$ | $\Delta T_3 = 0.08$ |
| | $\overline{\Delta T} = 0.04$ | | | $\overline{\Delta T} = 0.09$ | | |
| Structure Center | $\Delta T_1 = 0.12$ | $\Delta T_2 = 0.08$ | $\Delta T_3 = 0.16$ | $\Delta T_1 = 0.12$ | $\Delta T_2 = 0.08$ | $\Delta T_3 = 0.08$ |
| | $\overline{\Delta T} = 0.12$ | | | $\overline{\Delta T} = 0.09$ | | |
| Transmitted wave | $\Delta T_1 = -0.16$ | $\Delta T_2 = -0.12$ | $\Delta T_3 = -0.04$ | $\Delta T_1 = 0.00$ | $\Delta T_2 = 0.00$ | $\Delta T_3 = 0.00$ |
| | $\overline{\Delta T} = -0.11$ | | | $\overline{\Delta T} = 0.00$ | | |

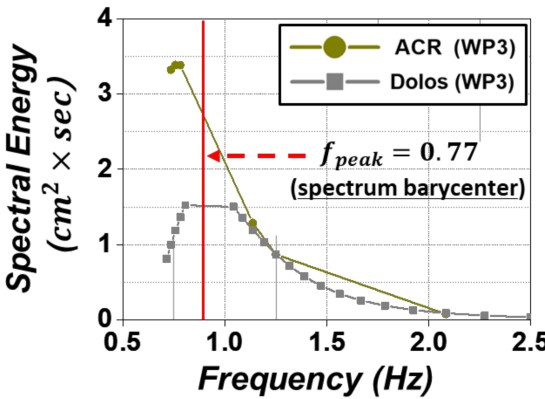

**Figure 9.** Movement of the spectrum barycenter.

### 3.2. Planar Current Movement and Current Vector Distribution Trends

To investigate the nearshore current behavior around the ACR, we performed a tracer experiment to determine the dispersion area and path of the dye around the dolos and ACR (Figure 10). Initially, the dispersion area of the dye was <1 m², and the spread of the dye was initiated by a nearshore current due to consecutive wave propagation inducing a water-level gradient around the ACR (Figure 10a,b). In both cases, aerial photographs revealed that a dominant current with a diagonal direction occurred particularly at the BO line, rather than at the BM or BS lines. Moreover, as the dispersion velocity of the dye increased continuously in the vicinity of the BO, the dispersion shapes were narrow and linear in both cases. However, we observed a significant difference in the current velocity between the dolos and ACR. The dispersion time of the ACR ($t/T_i = 15$) was almost 1.67-times longer than that of the dolos ($t/T_i = 9$), which reveals that the current velocity of the ACR was ~60% lower than that of the dolos (Figure 10g,h).

In general, when the porosity of the installed structure group increases, the kinetic energy transmitted to the rear side decreases under the same design and wave conditions [43]. The uniform installation of the dolos results in a short spatial distance between its structural units, whereas the ACR has sufficient void spaces owing to its porous shape, which reduces the current velocity. Moreover, as mentioned in a previous study [27], an increase in the wave height condition enlarges the wave setup. According to the results of the attenuation rate at the rear side of the structure, the ACR has a less variable attenuation rate ($R_A$) between the inner and outer domains compared to that of the dolos. For this reason, it is believed that the circular flow intensity of the ACR should be less than that of the dolos (Figure 10i,j), which corresponds to the results of the tracer experiment. These results emphasize the advantageous property of ACR in nearshore current mitigation.

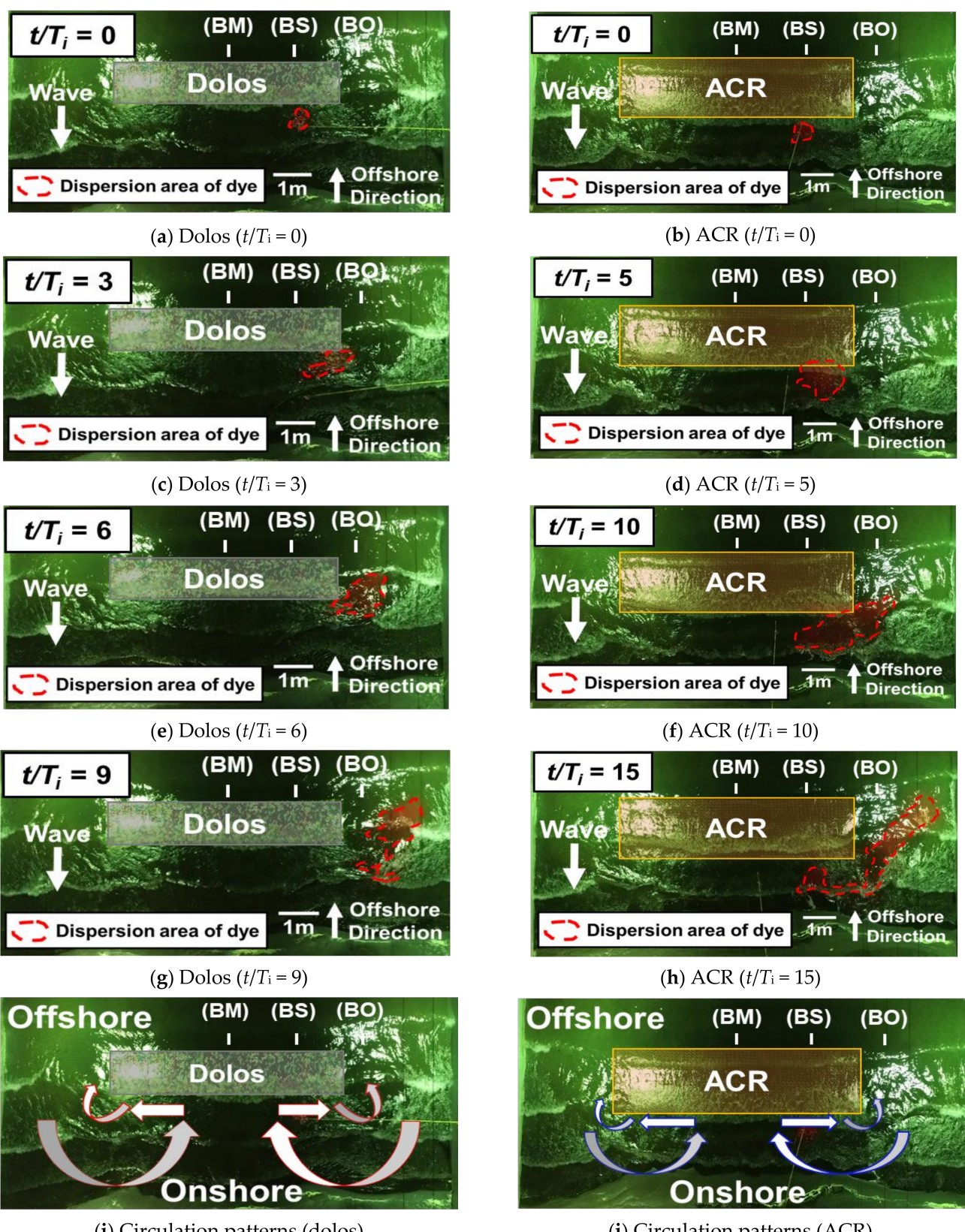

**Figure 10.** Planar movement of dye around the dolos (**a,c,g,e,i**) and ACR (**b,d,f,h,j**) at different time. The wave period (*T*i) was 1.3 s in both cases (BM: breakwater middle, BS: breakwater shoulder, and BO: breakwater open inlet).

To conduct a qualitative analysis of the current movement, we plotted the current vector. Figure 11 shows the results of the current vector distribution around both the dolos and ACR for the BM, BS, and BO lines.

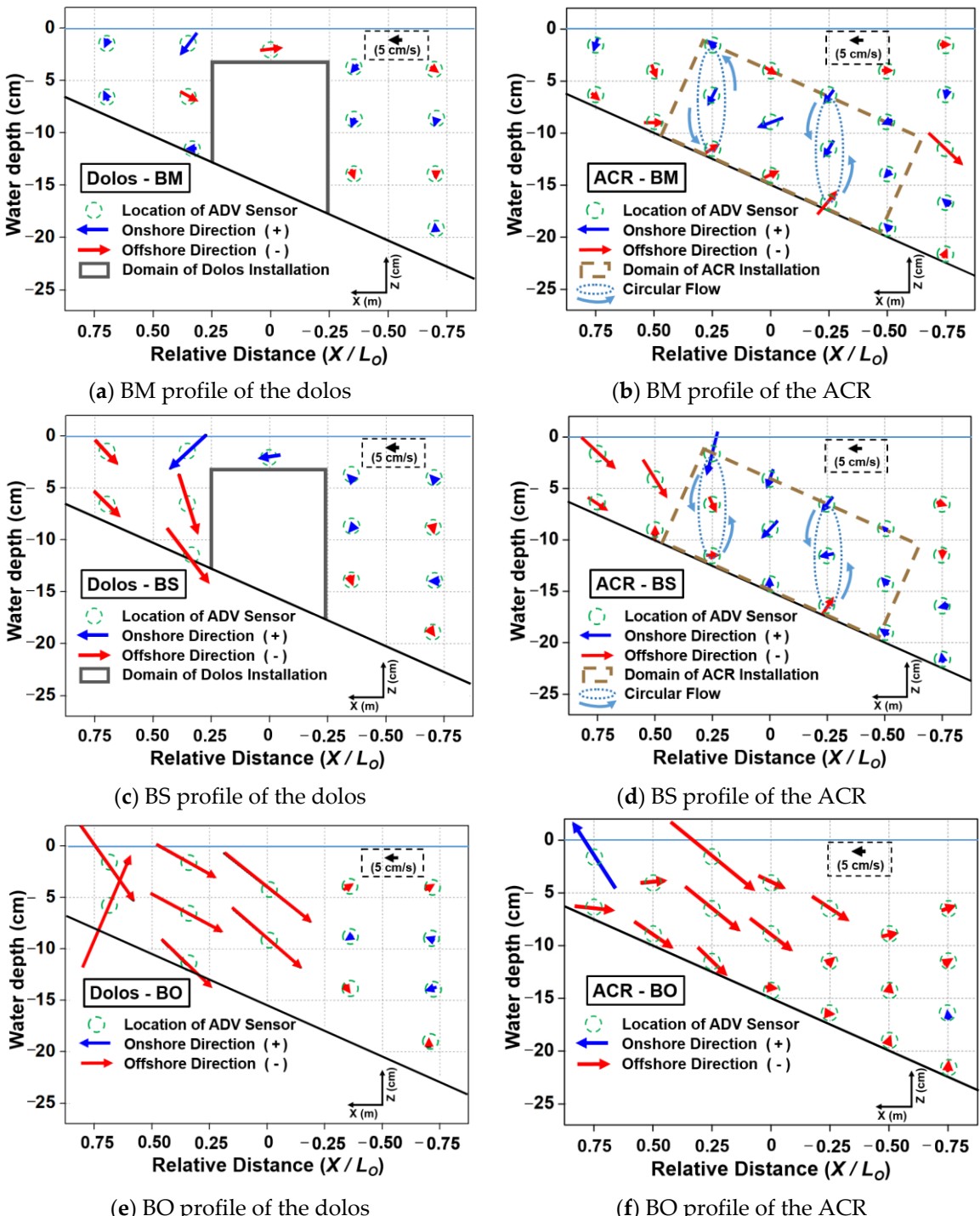

(**a**) BM profile of the dolos

(**b**) BM profile of the ACR

(**c**) BS profile of the dolos

(**d**) BS profile of the ACR

(**e**) BO profile of the dolos

(**f**) BO profile of the ACR

**Figure 11.** Distribution of current vectors around dolos (**a**,**c**,**e**) and ACR (**b**,**d**,**f**) (the vertical scale is exaggerated to enhance visibility).

First, in the BM line, the current intensity values of the dolos are slightly greater than those of the ACR at the onshore boundary of both structures (Figure 11a,b). In addition, in the internal ACR domain, a counterclockwise circular flow was observed, whereas the dolos did not allow any internal circular flow owing to the small porosity resulting from

its structural properties. We observed different current vector patterns for the BS line (Figure 11c,d). A strong current (~40 cm/s) occurred at the onshore boundary of the dolos seabed side, which can reduce the stability of the dolos. Moreover, the current values were larger in the dolos than those in the ACR domain. For the BO line, the currents in the offshore direction were dominant in both cases (Figure 11e,f). However, the maximum current vector magnitude of the dolos (53.5 cm/s) was 1.6-times greater than that of the ACR (33.3 cm/s), which caused a return flow around the structure boundary. Moreover, in the dolos case, rapid shrinkage of the fluid path due to its small porosity produced a return flow concentration, particularly at BS and BO, inducing the risk of intensive erosion with a strong undertow. In contrast, both the gradual crown depth variation and large voids between the wave and sand traps of the ACR allowed smooth water mass exchange through its structural unit. For this reason, the magnitude of the current vectors for dolos was greater than that of the ACR.

In summary, the current intensity of the ACR tends to be smaller than that of the dolos because of its structural porosity property; the ACR has the potential to prevent scour and erosion around its structure.

### 3.3. Topographical Variation Trends: Erosion and Deposition

Figure 12 shows the DoD results for the analysis of the topographical variation trends, which includes deposition and erosion. The longshore and wave directions were defined along the x and y axes, respectively. We observed both shoreline advancement and retreatment at the rear side of both the dolos and ACR structures.

At the point of sand deposition, the dolos exhibited swash zones of 0.7 m in the onshore direction and 0.3 m in the offshore direction from the original shoreline, as well as ~0.05 m of accumulated sediment. Interestingly, the ACR exhibited swash zones of 0.4 m in the onshore direction and 0.7 m in the offshore direction, as well as sand deposition with a height of ~0.1 m. The swash zone of the dolos was located closer to the onshore side than that of the ACR because the strong current passing over the dolos enhanced the longer wave run up to the onshore side.

Moreover, we found a diagonal deposition region from the rear side (from $X = 4.5$ m, $Y = 26.4$ m to $X = 8.5$ m, $Y = 26.4$ m) of the dolos to the outside boundary. In particular, severe erosion occurred on the rear side of the dolos around the BO plane. However, no apparent diagonal sedimentation or intensive erosion occurred on the rear side of the ACR.

In this regard, the dolos has a small void space between its armor unit and vertical cross section, which caused a reduction in the fluid path; however, the ACR had large void spaces between its structural units. Therefore, the porosity control of the ACR mitigated any severe erosion problems. We suggest that the cross-sectional design of the ACR prevents intensive erosion on the rear side of the structure.

Figure 13 shows the beach profiles of the dolos and ACR at the BM, BS, and BO for different wave generation times. In the BM profile, the shoreline accretion of the dolos and ACR were 30 and 71 cm, respectively. Moreover, the measured salient amplitudes were 150 and 191 cm at the dolos and ACR, respectively, which means that the ACR induced more sedimentation than that of the dolos (Figure 13c,d). In contrast, the dolos had a larger erosion volume than that of the ACR.

In the BS profile, the shoreline accretions of the dolos and ACR were 37 and 27 cm, respectively. The deposited area was formed from the shoreline to 2460 cm with a height of 3 cm in the dolos; however, ~3 cm of the deposited area was observed between 2640 and 2550 cm in the ACR (Figure 13e,f). In particular, severe scouring occurred at the rear side of the dolos in the BS line, causing intensive erosion (>10 cm), which induced the individual failure of the dolos armor units (Figure 13a). On the other hand, despite the slight topographical subsidence (~5 cm) that occurred around the onshore boundary, the ACR was maintained without any damage to the structural unit (Figure 13b). It is believed that both flexible material (HDPE) and movement as a single system significantly enhanced the structural stability of the ACR.

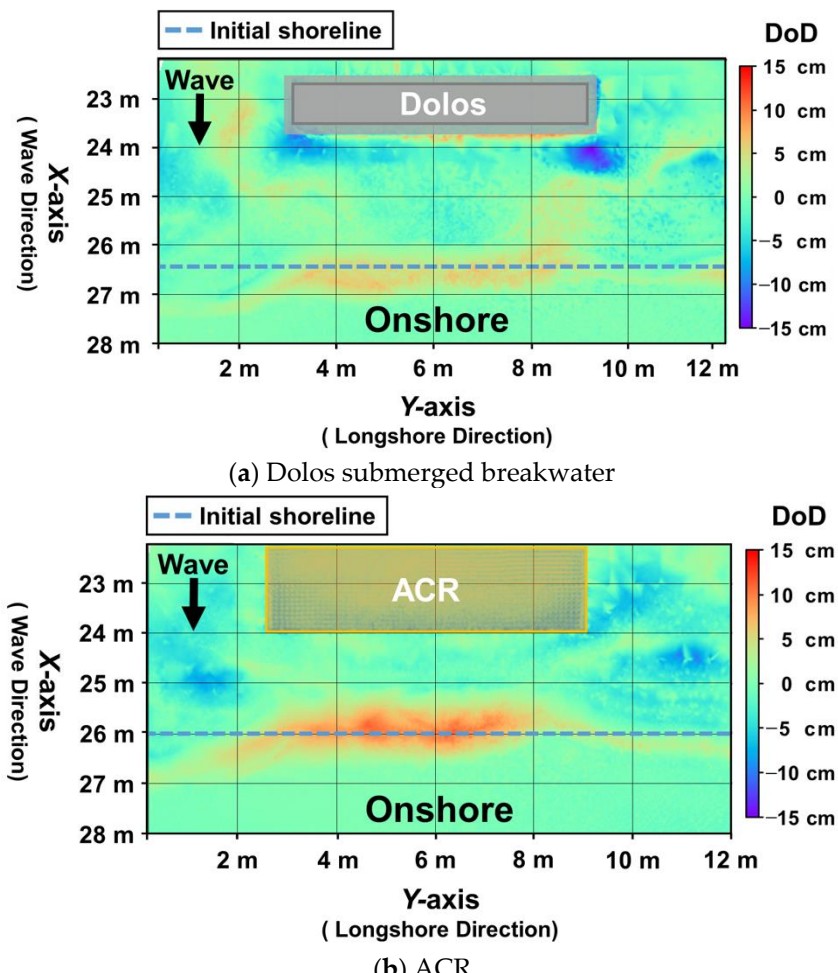

(**a**) Dolos submerged breakwater

(**b**) ACR

**Figure 12.** Results of the DoD computation of dolos (**a**) and ACR (**b**).

With regard to the shoreline change ($\Delta L$) on the BO line, the shoreline retreated 12 cm in the dolos case; however, the shoreline was stable in the ACR case (Figure 13g,h). In particular, the interaction between erosion and deposition formed a sandbar on the seabed. The height of the sandbar for the dolos was 4.8 cm at 2440 cm, and that of the ACR was 1.1 cm at 2420 cm. In addition, with laboratory experimental data, the scour position behind the offshore structure on the onshore side can be determined by Equation (9), as suggested by Young and Testik [44]. Based on Equation (9), the computed KC numbers of the dolos and ACR are 0.31 and 0.16, respectively, which satisfy the attached scour condition and support the results of the beach profile data on the BS line. In this respect, the topographical trends for both the dolos and ACR are reliable.

$$\begin{cases} KC = \left( \frac{H_i \pi}{W_{bw}} \right) \leq 3.14\,(\pi) : \textit{attached scour} \\ KC = \left( \frac{H_i \pi}{W_{bw}} \right) > 3.14\,(\pi) : \textit{detached scour} \end{cases} \tag{9}$$

where $KC$, $H_i$, and $w_{bw}$ represent the Keulegan–Carpenter number, incident wave height, and breakwater crest width, respectively.

In summary, the results clearly indicate that the ACR can lead to greater shoreline advancement behind the structure than the dolos. The beach-profile analysis quantitatively revealed the shoreline change ($\Delta L$) and salient amplitude ($Y_{\text{off}}$), which are the dominant parameters for a shoreline response analysis.

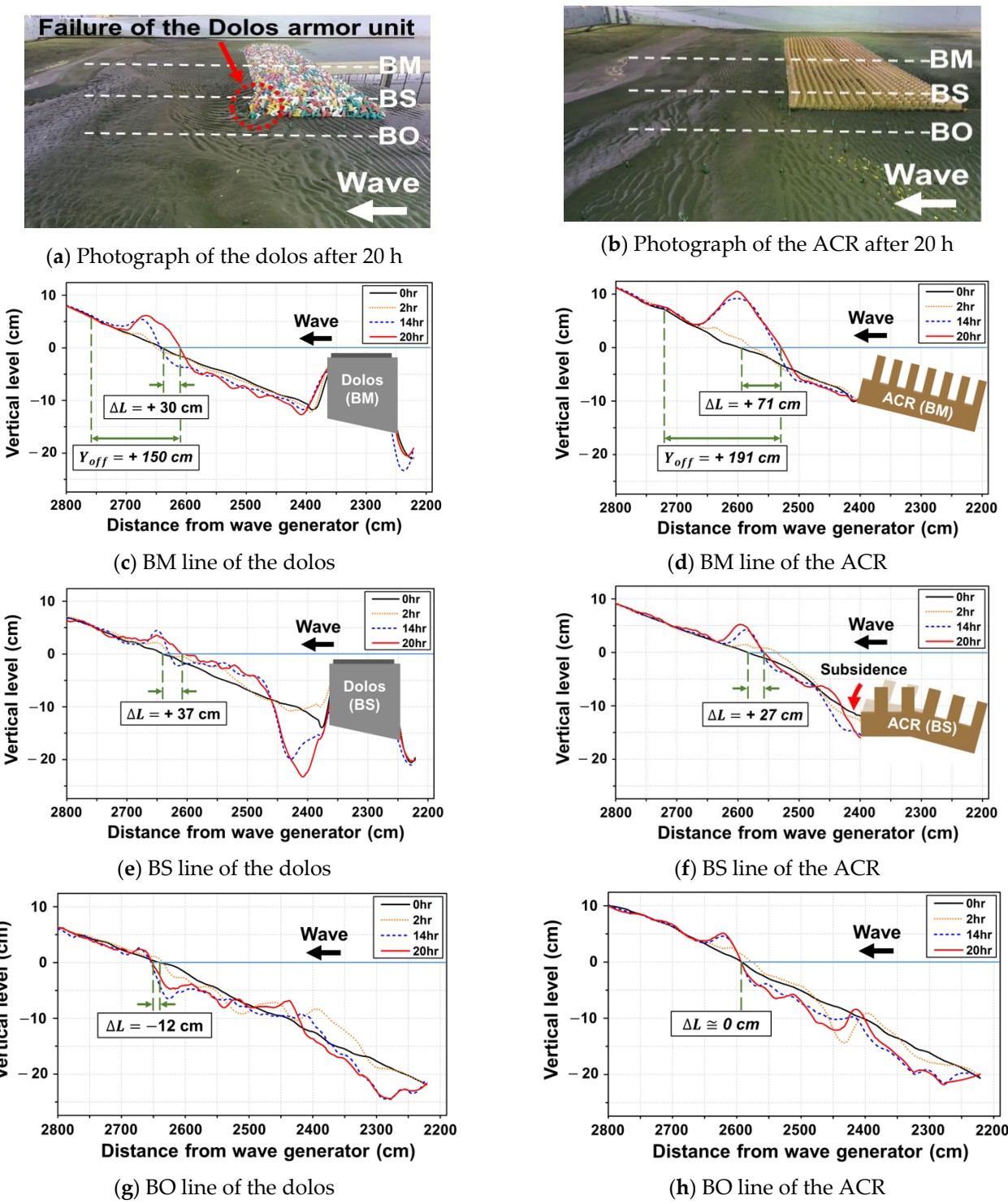

**Figure 13.** Vertical beach profiles around the dolos (**a,c,e,g**) and ACR (**b,d,f,h**) (the vertical scale is enlarged for visibility).

## 4. Conclusions

We performed a 3D large-scale experiment to determine the planar characteristics of an ACR on shoreline protection using various approaches. We demonstrated effective wave mitigation with the deployment of the ACR structure, which played a potential role in increasing coastal resilience. A wave profile analysis indicated that the ACR increased the nonlinearity of the wave, leading to effective wave breaking. The ACR exhibited a smaller

wave delay influence than that of the dolos, which is a positive feature for mitigating both wave diffraction and nearshore currents.

In tracer experiments, the current in the offshore direction was dominant, and the current intensity increased when it passed the BO. It is worth noting that the porous ACR structure contributed to mitigating the current intensity compared to that of the dolos structure. The quantitative results of the vertical beach profile indicate that the ACR tended to have a smaller current intensity (~60%) compared to those of the dolos cases, particularly at the onshore boundary of each structure, which prevents severe scouring around the structure.

The DoD profiles of the ACR represented a smaller eroded area at the rear side of the structure. Compared to the dolos case, a quantitative analysis of the beach profile revealed that the ACR induced greater shoreline advancement at the BM, and the eroded height was smaller at the BS. Moreover, at the BO, the shoreline retreated in the dolos case, whereas the coastline of the ACR case was stable.

These results indicate that the ACR played a positive role in shoreline stability in terms of both erosion and deposition. Based on these results, we expect that the ACR can be applied as a countermeasure structure to maintain shoreline stability and enhance coastal resilience.

In future studies, the authors aim to investigate the diffraction effects and average current vector trends in the longshore direction ($Y$ axis) of the ACR and provide more experimental data.

**Author Contributions:** Conceptualization, S.K. and S.H.; methodology, Y.K., J.L., A.P., W.K. and S.H.; formal analysis, S.H. and S.B.; literature review, S.H. and S.B.; writing—original draft preparation, S.H. and S.B.; writing—review and editing, S.K., J.L. and S.H.; visualization, S.H. and S.B.; supervision, A.P. and W.K.; project administration, S.K., J.L. and S.H. All authors have read and agreed to the published version of the manuscript.

**Funding:** This research was a part of the project titled 'Practical Technologies for Coastal Erosion Control and Countermeasure', funded by the Ministry of Oceans and Fisheries, Korea (20180404).

**Institutional Review Board Statement:** Not applicable.

**Informed Consent Statement:** Not applicable.

**Data Availability Statement:** No new data were created or analyzed in this study. Data sharing is not applicable to this article.

**Acknowledgments:** The authors acknowledge Dong Soo Hur of Gyeongsang National University for sharing his valuable knowledge and experience when reviewing the concepts and conditions of this experiment. Gwang Soo Lee, president of the Han Ocean Corp., provided an artificial coral reef structure that enabled us to conduct this experiment. We thank Dong Ha Lee of Kangwon National University who provided the LiDAR for this research. Additionally, we thank the members of the Balai Pantai Research Center in Bali, Indonesia. Huda Bachtiar performed many administrative works for this research. We thank Muhammad Hendro Setiawan and Ma'ruf Hadi Sutanto for discussions regarding the methods and experimental designs. We especially thank the laboratory technicians Nono Suparno, I. Ketut Heri Setiawan, Septian Setio Putro, and Kadek Pastika for their contributions to the experiments. Yolla Jessika Liauw, a researcher from Udayana University, is also appreciated. The authors would like to thank the editor and four reviewers for their valuable comments and constructive suggestions on this manuscript.

**Conflicts of Interest:** The authors declare no conflict of interest.

## Abbreviation and Symbols

| | |
|---|---|
| ACR | Artificial Coral Reef |
| $H_o$ | Deep-water wave height |
| $L_o$ | Wavelength |
| $D_{50}$ | Median grain size |
| $\Omega$ | Dean's parameter |
| $H_i$ | Incident wave height |
| $H$ | Wave height |
| $R_A$ | Attenuation rate |
| $T_i$ | Period of incident wave |
| WP | Wave Probe |
| $\eta$ | Wave profile |
| BM | Breakwater Middle |
| BS | Breakwater Shoulder |
| BO | Breakwater Open inlet |
| $u$ | Horizontal velocity |
| $\bar{u}$ | Average velocity in the horizontal direction ($x$-axis) |
| $w$ | Vertical velocity |
| $\bar{w}$ | Average velocity in the vertical direction ($z$-axis). |
| LiDAR | Light detection and ranging |
| DEM | Digital Elevation Model |
| DoD | Difference of DEM |
| $\Delta L$ | Shoreline change |
| $\Delta T$ | Wave delay parameter |
| $\overline{\Delta T}$ | Averaged wave delay parameter |
| KC | Keulegan-Carpenter number |
| $W_{bw}$ | Width of the breakwater crest |

## Appendix A. Table and Figures

**Table A1.** Wave attenuation rate ($R_A$) results.

| **Attenuation Rate of the Dolos** | | | | |
|---|---|---|---|---|
| *X/Lo* | **WP1** | **WP2** | **WP3** | **WP4** |
| −0.7 | 0.00 | 0.00 | 0.00 | 0.00 |
| −0.35 | 0.04 | 0.11 | 0.03 | −0.08 |
| 0 | 0.11 | −0.12 | −0.23 | 0.09 |
| 0.35 | 0.06 | 0.56 | 0.61 | −0.12 |
| 0.7 | 0.17 | 0.52 | 0.55 | 0.14 |
| **Attenuation Rate of the ACR** | | | | |
| *X/Lo* | **WP1** | **WP2** | **WP3** | **WP4** |
| −0.75 | 0.00 | 0.00 | 0.00 | 0.00 |
| −0.50 | −0.04 | −0.05 | −0.13 | −0.03 |
| −0.25 | 0.02 | −0.13 | −0.07 | −0.07 |
| 0 | 0.08 | −0.06 | 0.06 | 0.00 |
| 0.25 | 0.05 | 0.17 | 0.35 | −0.04 |
| 0.50 | 0.17 | 0.37 | 0.49 | 0.04 |
| 0.75 | 0.29 | 0.48 | 0.44 | 0.17 |

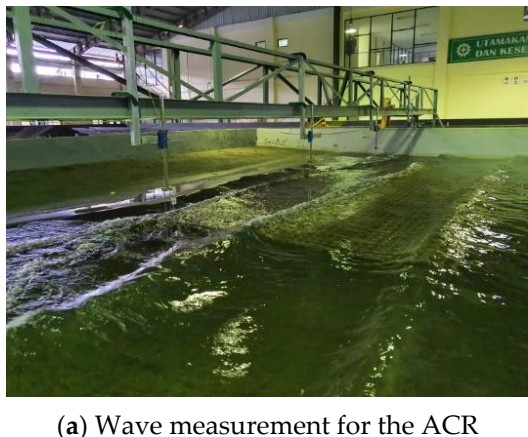

(**a**) Wave measurement for the ACR

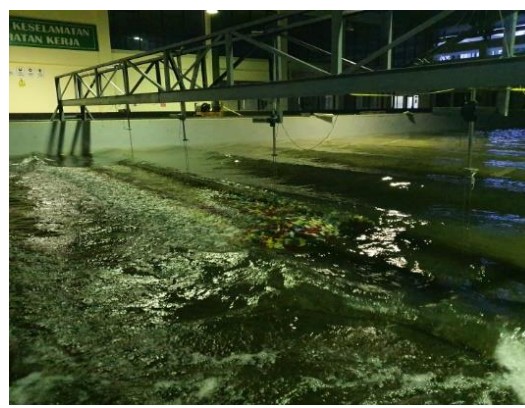

(**b**) Wave measurement for the dolos

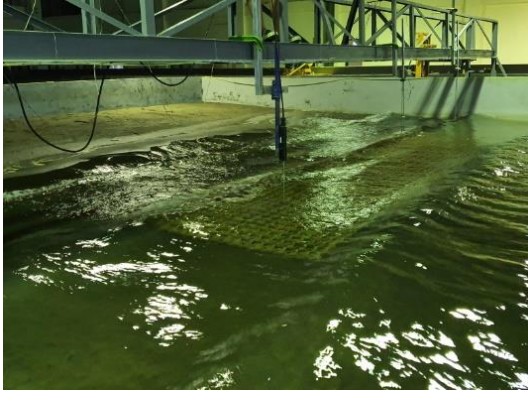

(**c**) Current measurement for the ACR

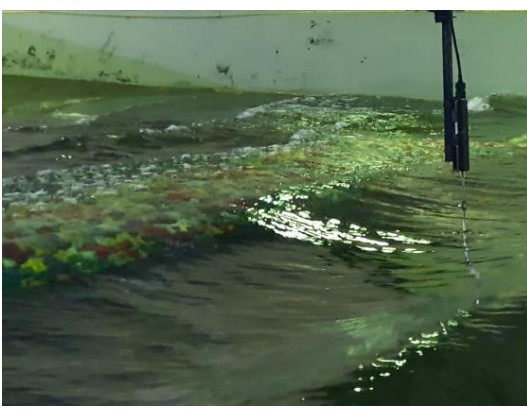

(**d**) Current measurement for the dolos

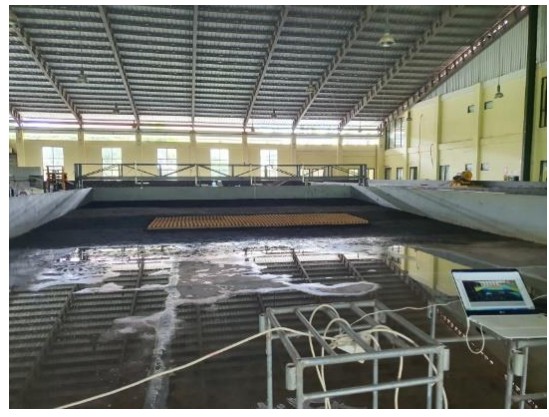

(**e**) LiDAR detection for the ACR

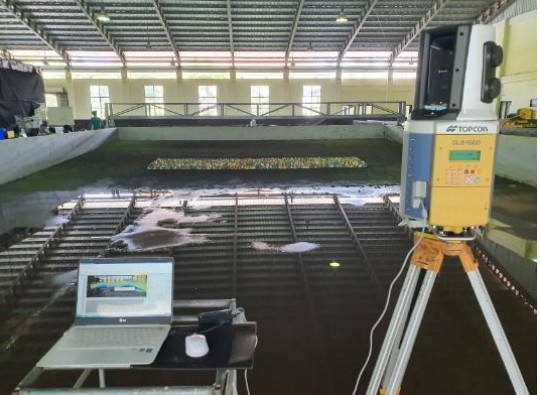

(**f**) LiDAR detection for the dolos

**Figure A1.** Photos of the model experiment around ACR (**a**,**c**,**e**) and dolos (**b**,**d**,**f**).

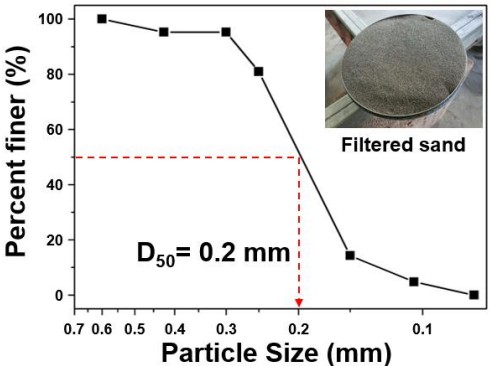

**Figure A2.** Particle-size distribution results.

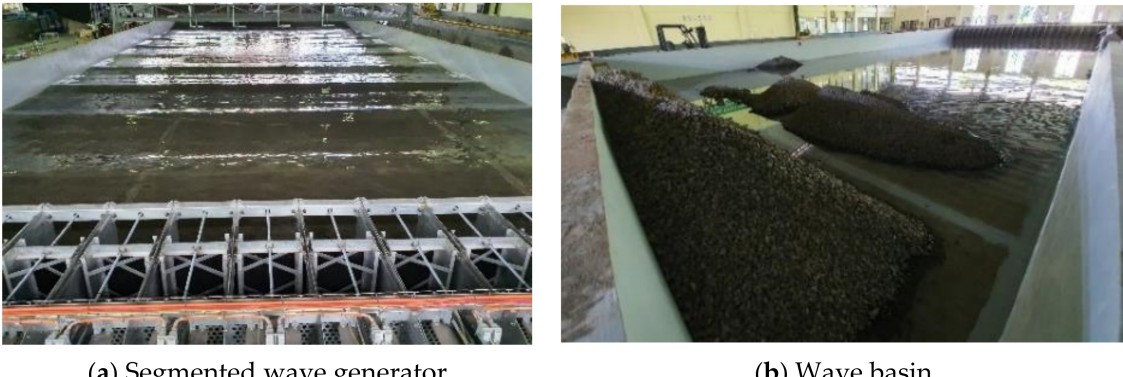

(**a**) Segmented wave generator                   (**b**) Wave basin

**Figure A3.** Experimental apparatus for the wave attenuation performance.

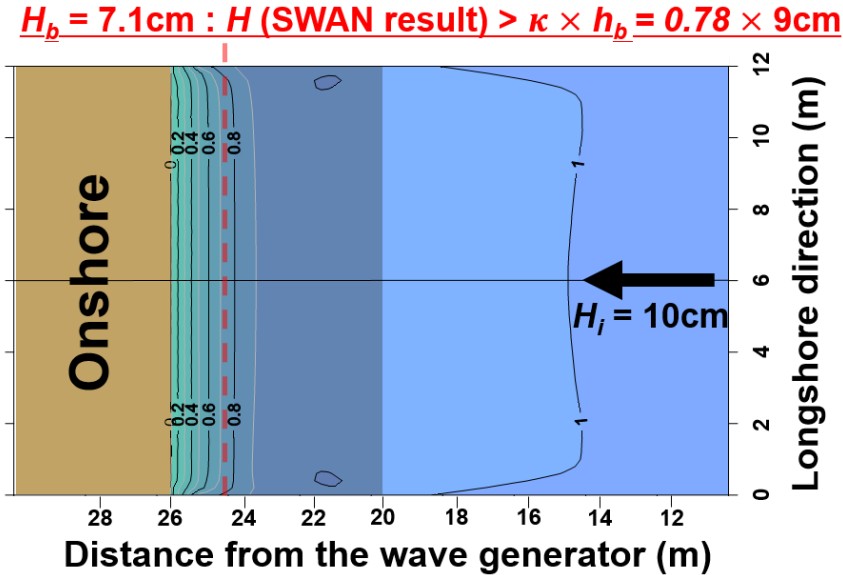

**Figure A4.** Determination of the height and depth of a breaking wave.

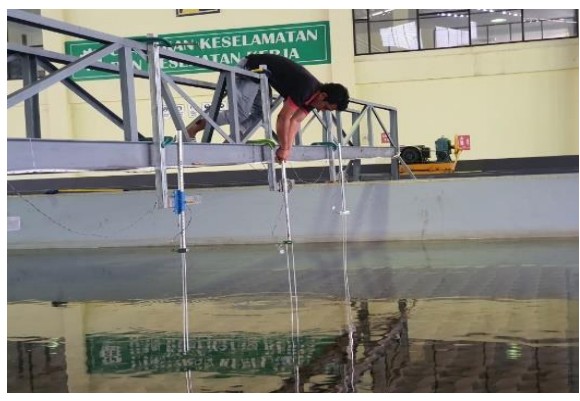
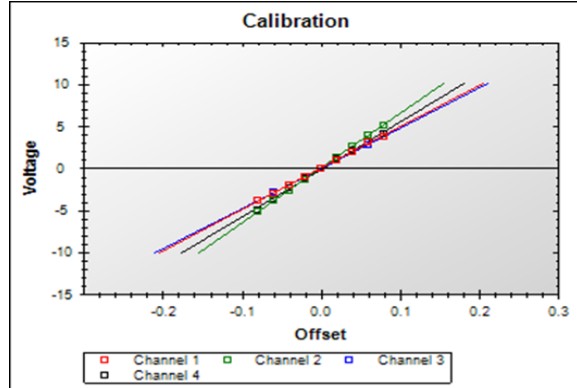

(**a**) Changing the wave probe location           (**b**) Regression line of the water level-voltage

**Figure A5.** Wave probe calibration (**a**) based on water level-voltage relation (**b**).

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
