# Peer review of "Planar Installation Characteristics of Crown Depth-Variable Artificial Coral Reef on Improving Coastal Resilience: A 3D Large-Scale Experiment"

_water, doi:10.3390/w13111526_

Round 1

Reviewer 1 Report

I appreciate your effort to deal with my comments. In many occasions, this effort is apparently insufficient, as acknowledged in your rebuttal.  In view of your intention to deal with some of those points in your further studies, I remove my objection to publication of the manuscript. Minor text editing however is still required. 

Author Response

Reply to Reviewer #1

We thank the reviewer 1 for specific suggestions and recommendations in the prior review. We have attempted to check English language style and minor spell check.

[General comment]

Reviewer #1: I appreciate your effort to deal with my comments. In many occasions, this effort is apparently insufficient, as acknowledged in your rebuttal.  In view of your intention to deal with some of those points in your further studies, I remove my objection to publication of the manuscript. Minor text editing however is still required. 

Reviewer 2 Report

The following are the review comments for the manuscript titled “Planar Installation Characteristics of Crown Depth-Variable Artificial Coral Reef on Improving Coastal Resilience: A 3D Large-scale Experiment”, submitted to MDPI-Water journal.

The authors presented and discussed the process for 3D laboratory simulation of both submerged Dolos and Artificial Coral Reef (ACR) breakwaters intending to investigate the achieved attenuation effects on the wave height, current velocity, score, and sedimentation on and around the ACR.

In general, the topic is worthy of investigation and of interest to the readers of ‘water’. The authors addressed the comments from a previous round of review, where, clearly the quality of the manuscript has improved significantly. However, there are still issues and questions that need to be discussed before a final publication.

Please see the below comments in two categories of major and minor points.

Major Points:

  1. I believe the structure of the Introduction section requires reconsideration. Either the title for sub-section “1.1. Developments of coastal erosion prevention method” is incomplete or the content does not clearly convey the intended plan to discuss the historical development aspects. Having said that, the existing content does not also read coherently, and there is no clear hierarchy/chronology in presenting the concepts, findings or approaches.
  2. I would also question where in the introduction the authors provided a literature review about "Improving Coastal Resilience" to then transition that to the concept of ACRs. In the introduction, the authors' previous works are not thoroughly and clearly discussed, to inform readers about how the concepts evolved over the course of years.
  3. In sub-section “1.2. Artificial coral reef (ACR) structures”, in lines 94-96, there is content about “wave trap” and “sand trap”; but they are not mentioned and used in any other sections.
  4. In Figure 1, the middle image is not clear. The majority of the important stuff in the picture is covered by text and annotations. The image on the left flows into the middle image; while there is no relationship between a site with natural coral reefs and a site with ACR. The issue with the image on the right is that it suggests there are three important crown depth values R1, R2, and R3; but in the other sections of the manuscript such an aspect is not discussed.
  5. The paragraph on the lines 102-109 is one of the most important introductory information. But the authors did not make use of that completely. Hence, there still remain questions as to what the background for ACR has been and what has already been done which are not discussed in this under-review manuscript.
  6. Following the above comments, at the end of the Introduction section, there is no clear conclusion as to what to expect in this under-review manuscript and why.
  7. For Eq. (1), how the scaling has happened for Hb=2.5m in reality and to 7.1cm in the lab? Is it related to what is explained in lines 136-137: "Moreover, we used 1.3s of wave period condition to calculate Dean’s parameter by considering Froude similarity scale (1:5) applied for this experiment"?
  8. It would be necessary to justify why Dolos armours are used as the one to compare with ACR. The first mention is in line 157.
  9. In lines 165-166, can the authors justify and explain their statement: “The length and time scale for model to prototype are set as 1:25 and 1:5, respectively”?
  10. There needs to be an explanation and justification as to why the stated dimensions in Figure 2 have been chosen. Given the chosen length scale, what would be the actual length of the Dolos or ACR on a real site? Furthermore, how would the authors justify the chosen 3m gap between the boundaries of the submerged Dolos or ACR and the boundaries of the testing facility?
  11. The summary presented in the last paragraph in section “3.1. Wave attenuation trends” is more about “wave mitigating effects”. A change needs to be considered.
  12. Considering Figure 10, please explain more why there is a strong undertow like current. There is a brief mention in the manuscript; but not through the lenses of discussion.
  13. Figure 12a shows failure of the Dolos units. Moreover, Figure 12f (and lines 385-386) shows/presents deformation of the ACR structure. Can you explain how flexible is the HDPE material? Or the plot in Figure 12f might be an exaggerated one?

Minor Points

There are a number of typographical, grammar and content issues. For the time being, a few of them are mentioned below:

  1. In line 151, “The beach is made of both …”, should be changed to “The actual beach is made of both …”.
  2. In line 98 and then 158, the terms “crown depth” and “crown depth (R)” are mentioned, respectively. But they have not introduced clearly, except a mention in Figure 1b.
  3. In line 163, it is said that “The water depth of the Dolos and ACR are 32 cm and 30 cm respectively." Can the water depths be shown in Figure 2? That should clarify a few matters; such as the 2cm water column above the Dolos units or Crown itself.
  4. Figure 2 is one of the very important and useful figures in the manuscript. Can the authors provide a different format with higher quality and larger images?
  5. Please provide the references for the statement in lines 175-177 regarding wave measurement in the lab.
  6. In line 191, for the first time, WP3 or WP4 are stated in-text. Please in a bracket refer the readers to Figure 3.
  7. In Figure 7, for the first time X/Lo is stated; but has not been introduced anywhere else.
  8. In line 249, “domain” needs to be changed to “the main”.
  9. In between lines 215 and 216, the mentioned equations have repeated equation numbers.
  10. In line 338, instead of “threat”, could endanger or jeopardize be a better word?
  11. In between lines 396 and 398, the mentioned equation has a repeated equation number. Having corrected it, what should be the referenced equation number in line 393?
  12. Figures A2 and A3 are not referred to within the manuscript. What are the benefits of having them included? or why were they perceived as less important to be included in the appendix?

Author Response

We thank the reviewer 2 for careful review of our paper and constructive suggestions. Please see our responses to your comments underneath each point. We believe these thorough comments and responses to the comments have made the paper stronger and more useful.  We do truly appreciate the time and effort of the reviewer 2 again. Comments are in italics and our responses are in boldface.

Reviewer 3 Report

The article, in se, is interesting and would deserve to be published. However, the English style is poor and the research is not properly organized in any parts. I suggest reconsidering the article after major revisions.

Detailed comments are attached 

Author Response

We thank the reviewer 3 for careful review of our paper and helpful suggestions. Please see our responses to your comments underneath each point. We believe these thorough comments and responses to the comments have made the paper stronger and more useful.  We do truly appreciate the time and effort of the reviewer 3 again. Comments are in italics and our responses are in boldface.

Reviewer 4 Report

3D large-scale lab experiments were conducted in this study. The authors analyzed the wave characteristics including wave attenuation trends, wave profile deformation by using Dolos and ACR and compared the results from both experiments. Current behaviors around ACR and around Dolos are also compared. Erosion and deposition caused by wave around Dolos and ACR are investigated. They come to the conclusion that ACR can facilitate the deposition and thus maintaining the shoreline stability from erosion. The analysis is thorough and the findings are practical for engineering. However, there are some aspects can be improved. This reviewer would suggest a minor revision. Please find my comments below:

Line 17: has significantly paid -à has been significantly paid

Line 24: may add a short and brief definition of Dolos submerged breakwater.

Line 37: what is a gained area? May need to replace another word, like deposition.

Line 41: the reason may also be land subsidence, for example coastal zone land lost along Norther Gulf of Mexico.

Line 46: what is full name of EUROSION?

Line 49-51: may add some literatures for those approaches.

Line 52-77: instead of listing the studies, I would suggest the authors to briefly summarize the results from these studies, what did they find or how does wave affect the land erosion, what are the differences of those studies compared with this paper, and what are the new methods or advantages of this paper compared with previous studies…

Line 157: may briefly introduce the Dolos submerged breakwater including its materials, dimensions, locations, and why the authors choose to use this specific breakwater and make comparison between Dolos and ACR?

Line 171: what is the vertical location of those wave gauges and wave probes?

Line 175: how did the authors calibrate the wave probe?

Line 184: the authors mentioned the time series water level data from offshore structure, but didn’t show it anywhere, I’d like to suggest either show the time series or remove these sentences.

Figure 4: I would suggest use same scale of x-axis for both subplot a and b, also, why did the authors use different distribution of Dolos and ACR (width of Dolos is different with that of ACR)?

The results and analysis are not well organized. I would suggest the authors present the results in the same order of that presented in the method session.

Author Response

We thank the reviewer 4 for careful review of our paper and specific comments. Please see our responses to your comments underneath each point. We believe these thorough comments and responses to the comments have made the paper stronger and more useful.  We do appreciate the time and effort of the reviewer 4 again. Comments are in italics and our responses are in boldface.

Round 2

Reviewer 2 Report

The following are the review comments for the revised version of the manuscript titled “Planar Installation Characteristics of Crown Depth-Variable Artificial Coral Reef on Improving Coastal Resilience: A 3D Large-scale Experiment”, submitted to MDPI-Water journal.

The authors presented and discussed the process for 3D laboratory simulation of both submerged Dolos and Artificial Coral Reef (ACR) breakwaters, as well as the achieved attenuation effects on the wave height, undertow current velocity, scour, and sedimentation on and around the ACR.

This reviewer thanks the authors for considering updates and editions to their manuscript. Perhaps, responding to different reviewers at the same time involved some challenges as there are texts with various font colour and background colour. Hence, in this current form, reading the content as a reviewer is not easy (but rather confusing).

Apart from this relatively easy-to-fix issue, the flow of the content is currently highly distorted which is unfortunately resonated by an extensive number of typographical and grammatical issues all around the manuscript. I believe all such issues should be resolved if the intention is to keep the readers engaged by the content.

Furthermore, considering the responses provide by the authors, there are still the following issues and questions that need to be discussed before a final publication:

  1. On the use and importance of the “sand trap” effect, please explain your observation during the laboratory simulation, as well as your prediction of the real-life applications, about the capacity of the sand traps, and the possibility of them being buried sometimes after installation of ACRs. That is, if the sand traps are already full and there remains no trapping possibility, the anticipated effects would never be materialized.
  2. For Eq. (1), still, it is not clear how Hb=7.1cm was calculated.
  3. There needs to be more explanation and justification about the chosen 3m gap between the boundaries of the submerged Dolos/ACR and the boundaries of the testing facility. Looking at Figure 11-e and f, Figure 12, and Figure 13-g and h, how the authors justify that there was no boundary effect?
  4. Figures A2 and A3 are not referred to within the manuscript, while Figures A1 and A4 are referred to in lines 163 and 232, respectively.
  5. Equation (9) does not have a reference.
  6. On line 215, there is an extra copy of Equation (2) from line 207.
  7. It is still unclear why there are 3 values of ΔT from equation (5) that can be averaged through Equation (6).

Reviewer 3 Report

I really appreciate the effort the authors profused in revising their manuscript. I suggest it be accepted for publication

Author Response

We appreciate the reviewer 3 for helpful suggestions with relevant research papers. We believe these thorough comments and responses to the comments have made the paper stronger and more useful.  We do truly appreciate the time and effort of the reviewer 3 again.

[General comment]

Reviewer #3: I really appreciate the effort the authors profused in revising their manuscript. I suggest it be accepted for publication

Reviewer 4 Report

I am satisfied with the current manuscript. All my concerns and questions have been properly addressed. I believe it has been sufficiently improved, therefore, I suggest it to be sent to publication.

Author Response

We thank the reviewer 4 for careful review of our paper and specific comments including better scientific expression. We believe these thorough comments and responses to the comments have made the paper stronger and more useful.  We do appreciate the time and effort of the reviewer 4 again.

[General comment]

Reviewer #4: I am satisfied with the current manuscript. All my concerns and questions have been properly addressed. I believe it has been sufficiently improved, therefore, I suggest it to be sent to publication.

Round 3

Reviewer 2 Report

Firstly, the authors' determination to make improvements is acknowledged. However, this reviewer believes that there is a lot of room to correct the grammatical issues and prove the readability. In between the new changes, in lines 266-267 the reference should be to Figure 4 on line 268. I wish luck in your research, and look forward to reading more of your works.

Author Response

This manuscript is a resubmission of an earlier submission. The following is a list of the peer review reports and author responses from that submission.

Round 1

Reviewer 1 Report

A 3D large-scale experiment has been carried out to determine the planar effects of an ACR (artificial coral reef) on the mitigation of the coastal erosion with various spatial analyses. They concluded that agreement between the experimental results and the empirical equation for the shoreline response indicates that using this equation for predicting the shoreline response is reasonable and practical. Although the paper contains interesting results, major revisions are required especially for interpretation and formulation of the shoreline change.

  1. In “3.7 Shoreline analysis using empirical equation”, the way of writing is not suitable. In Figure 12, they say in the legend “Reef data”, but this is totally wrong. The plotted data are from combined island and reef of B&A. The reviewer strongly suggests the authors to be strict for citation, and to plot only reef data in this diagram.
  2. In the manuscript, nothing has been described for the inherent similitude problem of sediment movement phenomena. Although B&A (2001)’s empirical formula has been proposed based on field data, it can be applied to the laboratory experiment? There is no problem related to similitude? Moreover, it is highly well known that the critical value demarcating erosional and accretional profile proposed by Sunamura & Horikawa’s is totally different under field conditions, not simply Cs=8. Like this, it is highly important to separate interpretations and discussions into laboratory scale and field site scale phenomena.

Reviewer 2 Report

The manuscript describes results of an experimental study aimed at assessing the performance of Artificial Coral Reef (ARC) Structure in enhancing coastal resilience. The experiments are performed in a wave basin and an attempt is made to carry out comparison with results obtained in the same facility using a dolos. The performance of the device is judged based on the wave attenuation and the transmission coefficient, as well as by analyzing the shoreline response.  Wave-induced currents were measured using an ADV.  The conclusion reached in this study is that ARC is advantageous to dolos. The study clearly indicates that application of ARC to enhance coastal resilience in certain cases may indeed be preferable to other options, and for that reason, the paper may be eventually recommended for publications. However, in the present form it leaves too many questions unanswered. Those question should be addressed in the revision.

The scaling applied in the present study should be further clarified. The experiments are carried out for a single very steep wave amplitude at a single incident wavelength. The effect of the ACR is estimated in part by comparison of measurements with those carried out in the presence of a conventional submerged breakwater (dolos). The operational conditions at which the two devices were tested, as well as the dimensions of the dolos and of the ACR are quite different. It is not specified how exactly the comparison is carried out. Moreover, for an ARC that has a complex geometry and therefore multiple characteristic length scales, it remains unclear how experiments at a single wavelength can be interpreted for incident waves that have a broad spectrum of frequencies. The authors justify their use of steep waves by the need to induce erosion. However, this approach does not allow long term assessment of beach erosion under action of waves that are not that steep where nevertheless the beach may undergo erosion but nonlinear effects are less pronounced.  It is noted in line 219 that higher waves broke earlier (as can be expected). However, no quantitative results on experiments with different wave heights are presented. In general, the wave conditions in the experiments can probably be specified better; this will allow some estimate of the limits of their validity.

The experiments are not described in sufficient detail. While the spanwise locations are specified sufficiently well, it remains unclear where exactly the sensors in the wave propagation direction were positioned (the only information available appears in Fig. 8a). Since the duration of measurements is quite long and as Fig. 4 clearly demonstrates, the incident wave is far from being monochromatic, a variety of resonant standing waves can be excited in the limited-size facility; those waves, if they exist, can strongly affect the measurements of the reflection coefficient. The distance between the sensors also affects the wave phase measurements. In any case, it is not explained why wave phases are essential. Although they are used in discussion following Fig. 9, this discussion should be improved and the change of the wave propagation direction due to diffraction in the presence of finite length breakwater as well as of wavelength over varying depth should be discussed. This comment also applies to characterization of currents.

The digital elevation model (DEM) mentioned in line 187 should be described in some detail. The discussion of results of current measurements and of dual structure of ACR in Section 3.3 should be significantly improved. It is suggested that current around ACR should be mitigated to prevent erosion (lines 289-290), but there is no hint as to how this can be implemented in practice.

In Section 4 the apparent advantages of ACR over dolos are summarized. While it is quite possible that ACR may be preferable to other submerged breakwaters at least in some conditions, in view of the comments presented above and some additional points not raised here, the conclusions drawn in Section 4 seem not to be substantiated well enough.